# Pre-Diagnostic Circulating Resistin Concentrations Are Not Associated with Colorectal Cancer Risk in the European Prospective Investigation into Cancer and Nutrition Study

**DOI:** 10.3390/cancers14225499

**Published:** 2022-11-09

**Authors:** Thu-Thi Pham, Katharina Nimptsch, Krasimira Aleksandrova, Mazda Jenab, Robin Reichmann, Kana Wu, Anne Tjønneland, Cecilie Kyrø, Matthias B. Schulze, Rudolf Kaaks, Verena Katzke, Domenico Palli, Fabrizio Pasanisi, Fulvio Ricceri, Rosario Tumino, Vittorio Krogh, Jeanine Roodhart, Jesús Castilla, Maria-Jose Sánchez, Sandra Milena Colorado-Yohar, Justin Harbs, Martin Rutegård, Keren Papier, Elom K. Aglago, Niki Dimou, Ana-Lucia Mayen-Chacon, Elisabete Weiderpass, Tobias Pischon

**Affiliations:** 1Molecular Epidemiology Research Group, Max-Delbrueck-Center for Molecular Medicine in the Helmholtz Association (MDC), 13125 Berlin, Germany; 2Charité-Universitätsmedizin Berlin, Corporate Member of Freie Universität Berlin and Humboldt-Universität zu Berlin, 10117 Berlin, Germany; 3Department of Epidemiological Methods and Etiological Research, Leibniz Institute for Prevention Research and Epidemiology—BIPS, 28359 Bremen, Germany; 4Faculty of Human and Health Sciences, University of Bremen, 28359 Bremen, Germany; 5Nutrition and Metabolism Branch, International Agency for Research on Cancer (IARC-WHO), World Health Organization, 150 Cours Albert Thomas, CEDEX 08, 69372 Lyon, France; 6Department of Nutrition, Harvard T.H. Chan School of Public Health, Boston, MA 02115, USA; 7Danish Cancer Society Research Center, 2100 Copenhagen, Denmark; 8Department of Public Health, University of Copenhagen, DK-1353 Copenhagen, Denmark; 9Department of Molecular Epidemiology, German Institute of Human Nutrition Potsdam-Rehbruecke, 14558 Nuthetal, Germany; 10Institute of Nutritional Science, University of Potsdam, 14558 Nuthetal, Germany; 11Department of Cancer Epidemiology, German Cancer Research Center (DKFZ), 69120 Heidelberg, Germany; 12Cancer Risk Factors and Life-Style Epidemiology Unit, Institute for Cancer Research, Prevention and Clinical Network (ISPRO), 50139 Florence, Italy; 13Dipartimento di Medicina Clinica E Chirurgia, Federico Ii University, 80131 Naples, Italy; 14Centre for Biostatistics, Epidemiology, and Public Health (C-BEPH), Department of Clinical and Biological Sciences, University of Turin, 10043 Orbassano, Italy; 15Unit of Epidemiology, Regional Health Service ASL TO3, 10095 Grugliasco, Italy; 16Hyblean Association for Epidemiological Research, AIRE ONLUS, 97100 Ragusa, Italy; 17Epidemiology and Prevention Unit, Fondazione IRCCS Istituto Nazionale dei Tumori di Milano, Via Venezian 1, 20133 Milan, Italy; 18Department of Medical Oncology, UMC Utrecht, 3584 CX Utrecht, The Netherlands; 19Navarra Public Health Institute—IdiSNA, 31003 Pamplona, Spain; 20CIBER Epidemiología y Salud Pública (CIBERESP), 28029 Madrid, Spain; 21Escuela Andaluza de Salud Pública (EASP), 18011 Granada, Spain; 22Instituto de Investigación Biosanitaria ibs.GRANADA, 18012 Granada, Spain; 23Department of Preventive Medicine and Public Health, University of Granada, 18071 Granada, Spain; 24Department of Epidemiology, Murcia Regional Health Council, IMIB-Arrixaca, 30008 Murcia, Spain; 25Research Group on Demography and Health, National Faculty of Public Health, University of Antioquia, Medellín 050010, Colombia; 26Department of Radiation Sciences, Oncology, Umeå University, SE-901 87 Umeå, Sweden; 27Department of Surgical and Perioperative Sciences, Surgery, Umeå University, SE-901 87 Umeå, Sweden; 28Wallenberg Centre for Molecular Medicine, Umeå University, SE-901 87 Umeå, Sweden; 29Cancer Epidemiology Unit, Nuffield Department of Population Health, University of Oxford, Oxford OX3 7LF, UK; 30Department of Epidemiology and Biostatistics, Imperial College London, London W2 1PG, UK; 31International Agency for Research on Cancer, World Health Organization, 69372 Lyon, France; 32Max-Delbrueck-Center for Molecular Medicine in the Helmholtz Association (MDC), Biobank Technology Platform, 13125 Berlin, Germany; 33Berlin Institute of Health, Charité-Universitätsmedizin Berlin, Core Facility Biobank, 13125 Berlin, Germany

**Keywords:** pre-diagnostic resistin, colorectal cancer, risk, prospective, inflammation

## Abstract

**Simple Summary:**

Resistin has been proposed to link to cancer development via inflammatory processes. Prior case-control studies suggest higher post-diagnosis resistin concentrations in CRC cases compared to controls. Here, we found no association between pre-diagnostic circulating resistin concentrations and the risk of CRC; however, we observed a marginally significant association among cases (and their matched controls) diagnosed with CRC within the first two years of follow-up, whereas no such association was observed among cases (and their matched controls) diagnosed with CRC after two years of follow-up. We speculate that resistin is more likely a marker of existing tumors than a risk factor of CRC.

**Abstract:**

Resistin is a polypeptide implicated in inflammatory processes, and as such could be linked to colorectal carcinogenesis. In case-control studies, higher resistin levels have been found in colorectal cancer (CRC) patients compared to healthy individuals. However, evidence for the association between pre-diagnostic resistin and CRC risk is scarce. We investigated pre-diagnostic resistin concentrations and CRC risk within the European Prospective Investigation into Cancer and Nutrition using a nested case-control study among 1293 incident CRC-diagnosed cases and 1293 incidence density-matched controls. Conditional logistic regression models controlled for matching factors (age, sex, study center, fasting status, and women-related factors in women) and potential confounders (education, dietary and lifestyle factors, body mass index (BMI), BMI-adjusted waist circumference residuals) were used to estimate relative risks (RRs) and 95% confidence intervals (CIs) for CRC. Higher circulating resistin concentrations were not associated with CRC (RR per doubling resistin, 1.11; 95% CI 0.94–1.30; *p* = 0.22). There were also no associations with CRC subgroups defined by tumor subsite or sex. However, resistin was marginally associated with a higher CRC risk among participants followed-up maximally two years, but not among those followed-up after more than two years. We observed no substantial correlation between baseline circulating resistin concentrations and adiposity measures (BMI, waist circumference), adipokines (adiponectin, leptin), or metabolic and inflammatory biomarkers (C-reactive protein, C-peptide, high-density lipoprotein cholesterol, reactive oxygen metabolites) among controls. In this large-scale prospective cohort, there was little evidence of an association between baseline circulating resistin concentrations and CRC risk in European men and women.

## 1. Introduction

Resistin is a polypeptide consisting of 108 amino acids named after “resistance to insulin” that belongs to the “resistin-like molecules” family [1]. It was initially reported in rodents as a protein primarily secreted by adipocytes and plays a role in obesity-induced insulin resistance. Higher resistin concentrations have been found in obese as compared to non-obese animal models [1]. In humans, reports on the correlation between circulating resistin concentrations and adipose tissue mass have been inconsistent [2,3,4,5,6], and resistin was found to be predominantly produced by macrophages and monocytes, rather than adipocytes [2,3]. The production and upregulation of resistin occur during monocyte–macrophage differentiation [3] and resistin then promotes an M1-like (pro-inflammatory) phenotype in macrophages [7]. Increases in adipose tissue mass in humans may be accompanied by infiltration of macrophages and monocytes, which release resistin and may thereby affect multiple cell types and tissues [3,8]. Human resistin is relevant for inflammatory processes [8,9]. By various mechanisms, resistin receptor binding may lead to an upregulation of inflammatory cytokines, including interleukin-6 (IL-6) and tumor necrosis factor α (TNF-α) [8,9,10], to promote NF-kB activation [7], whereas resistin expression is also upregulated in the presence of inflammatory cytokines by a positive feedback loop, which may result in a vicious cycle and perpetuate inflammatory conditions [7,10,11]. Other than inflammation, resistin could promote the expression of adhesion molecules and growth factors that may promote angiogenesis [8,9]. Thus, resistin emerges as an important marker implicated in the development of cancer [8].

Colorectal cancer (CRC) is the third most common and second most fatal cancer worldwide [12]. The global burden of CRC is expected to increase by 60% by 2030 [13]. While European, Australia/New Zealand, and Northern American regions have both the highest CRC cancer incidence and mortality rate, the majority regions of Africa and South Central Asia have the lowest incidence rate [12]. Thus, CRC incidence has been suggested to correlate with socioeconomic development and may be linked to western lifestyles [12,13]. Furthermore, incidence of CRC is closely related to multiple factors, including a family history of colon polyps or inflammatory bowel diseases, socioeconomic status, lifestyle and dietary factors, and the gut microbiome, which shares the underlying mechanism related to inflammation, angiogenesis, and insulin resistance [14,15].

As both inflammation and angiogenesis are related to CRC, it was speculated whether resistin may also be relevant to the risk of CRC [8,16]. In fact, higher resistin levels have been found in CRC patients compared to healthy individuals in case-control studies [17]. However, it is unclear whether higher resistin levels are a cause or consequence of CRC development. Prospective studies with pre-diagnostic resistin concentrations are necessary to investigate whether higher resistin concentrations are associated with a higher risk of developing CRC. To date, there has been only one prospective study using data from 1224 postmenopausal women (427 CRC cases) in the Women’s Health Initiative (WHI), which found no significant associations between resistin and CRC risk [5]. However, different associations between inflammation and the risk of CRC have been observed in men versus women [18], suggesting that the association between resistin concentrations and the risk of CRC may differ by sex. Furthermore, that study did not separate the association by tumor subsite. Given that inflammation is more strongly related to colon cancer than to rectal cancer [19], an examination by tumor subsite is important to better understand the impact of resistin as an exposure.

Therefore, we conducted a nested case-control study within the European Prospective Investigation into Cancer and Nutrition (EPIC) study to investigate the association between pre-diagnostic resistin concentrations and CRC risk and to examine whether the association differed by cancer subsite or sex.

## 2. Materials and Methods

### 2.1. The European Prospective Investigation into Cancer and Nutrition Study

#### 2.1.1. Study Population

The EPIC study is a large, ongoing cohort study, details of which have been extensively reported elsewhere [20]. Briefly, the EPIC study was initiated in 1990 with the collaboration of 23 centers in 10 European countries, including Denmark, France, Germany, Greece, Italy, The Netherlands, Norway, Spain, Sweden, and the United Kingdom. Study participants’ enrollment in all EPIC centers had been completed in 2000, with 519,978 participants aged 35 to 70 years. In the current nested case-control study, data from Greece were not included due to administrative reasons.

#### 2.1.2. Assessments of Anthropometry, Lifestyle Factors, and Dietary Exposure

Weight and height were measured for all participants in all EPIC centers by trained observers, except in France, Oxford, and Norway [21]. In France and Oxford, weight and height were measured in a part of the population, whereas self-reported weight and height were available for all participants. For part of the Oxford cohort, linear regression models were used to predict sex- and age-specific values from individuals with both measured and self-reported body measures. For France, only the measured values were used. In Norway, only self-reported weight and height data were available. Waist circumference was measured in all centers except in Norway and Umeå, Sweden.

Lifestyle characteristics were collected via questionnaires at recruitment including questions on education, tobacco smoking status, consumption of alcoholic beverages, physical activity according to the validated Cambridge physical activity index, and for women, menstrual status, use of contraception, and hormone replacement therapy [20]. The Cambridge physical activity index describes four levels of physical activity (inactive, moderately inactive, moderately active, and active) based on recreational activity time in occupation, cycling, and other physical activities [22]. Questionnaires on lifestyle variables were developed and used independently in Denmark, Sweden, Norway, and the Naples center in Italy, while they were previously standardized in other countries. All lifestyle variable codes from different questionnaires were standardized to the core EPIC lifestyle questions using a comprehensive recoding scheme afterward. Baseline dietary exposure (including, among others, consumption of red meat, processed meat, dietary fiber, fruit, vegetable, dairy, fish, and shellfish intake, as well as alcohol consumption) was assessed on all EPIC participants by using locally adopted instruments, including food frequency questionnaires, diet history logs, and a combined method, and was used to estimate long-term usual dietary intake.

#### 2.1.3. Blood Collection

Biological samples including plasma, serum, leukocytes, and erythrocytes were collected at baseline from most participants, with the exception of those in France, the UK, Bilthoven (The Netherlands), and Norway, where only a proportion of the participants were invited for blood sampling [20]. For most EPIC centers, half of the blood samples were stored locally, and half were transported to the central repository of the International Agency for Research on Cancer (IARC) to be stored in straws in the vapor phase of liquid nitrogen at −196 °C. Blood samples were all stored locally in freezers at −70 °C in Sweden and nitrogen vapor at −150 °C in Denmark.

#### 2.1.4. Follow-Up for Cancer Incidence

Incident CRC cases were identified via regional cancer registries in Denmark, Italy, The Netherlands, Norway, Spain, Sweden, and the United Kingdom and by a combination of methods in France and Germany. The combination of methods included a direct follow-up procedure through study participants or their next of kin, and confirmation of tumors from a review of health insurance records and pathology registries. For centers that applied a combination of methods, individuals’ end of follow-up was considered the last known contact, date of diagnosis, or date of death, whichever came first.

### 2.2. The Nested Case-Control Study

#### 2.2.1. Study Design

Incident CRC cases were defined as EPIC participants who developed CRC after recruitment and before the closure dates. CRC was defined as a combination of tumors of the colon (the 10th Revision of the International Classification of Diseases (ICD-10) codes C18.0–C18.7), tumors that were overlapping or unspecified (C18.8–C18.9), and tumors of the rectum (C19–C20). The closure dates for the nested case-control study of the present analysis ranged from December 2001 to December 2005.

The control selection process was based on an incidence density sampling approach. One control was selected for each case from among a sample of those who were at risk at the time of diagnosis of the index case with available blood samples and was matched (1:1) by recruitment center, sex, age at recruitment (±2 years), date of blood collection (±3 months), time of day of blood collection (±4 h), fasting status at blood collection (<3, 3–6, and >6 h), and menopausal status (premenopausal, postmenopausal, perimenopausal, or surgically postmenopausal) for women. Premenopausal women were matched on phases of menstrual cycles and use of oral contraceptives, and postmenopausal women were matched on current hormonal replacement therapy use.

#### 2.2.2. Laboratory Analysis

Serum resistin concentrations were measured using human resistin ELISA assays (BioVendor Laboratory Medicine, Inc.; Brno, Czech Republic). Measurements were performed according to the manufacturer’s protocols. The mean inter-assay coefficients of variation of the laboratory analysis were 7.4% and 6.6% for quality-control high concentrations (18.3 ng/mL) and low concentrations (5.01 ng/mL), respectively. The mean inter-assay coefficient of variation was <10.4% for all pooled serum quality controls. Measurements and inter-assay coefficients of variation in adiponectin and high-molecular weight (HMW) adiponectin [23], leptin [24], soluble leptin receptor [24], reactive oxygen metabolites (ROM) [25], high-density lipoprotein cholesterol (HDL-C), C-peptide [26], high-sensitivity C-reactive protein (hsCRP) [27], and glycated hemoglobin A1c (Hba1c) [28] have been described previously.

#### 2.2.3. Final Dataset, Handling of Missing Data

The current analysis was based on data from participants in the EPIC CRC nested case-control study, for which plasma samples were available for laboratory analysis. Resistin concentrations were successfully analyzed in 1383 CRC cases and 1375 controls. After excluding participants without their matching pairs, a total of 1293 first-incident CRC cases and their matched controls were included in the current study.

Waist circumference measurements were missing in 77 matched case sets (6.0%), including 16 and 61 case sets from Norway and Umeå-Sweden, respectively. For 4 CRC cases, data on alcohol consumption, energy intake, and consumption of red meat, processed meat, dietary fiber, fruit, vegetable, dairy, fish, and shellfish intake were missing. Data for other variables were missing in cases and controls as follows: fasting status (20/21), smoking status (14/15), highest education level (43/34), physical activity index (16/18), and diabetes (73/90). We checked the distribution of resistin concentrations in participants with missing and no missing data on these variables and observed no differences. Together with the arbitrary missing data patterns generated by the PROC MI procedure in SAS^®^ Enterprise Guide^®^ 8.3 (SAS Institute Inc., Cary, NC, USA), we assumed that the data were missing completely at random. Hence, missing data were imputed by sex-specific medians for discrete variables and sex-specific modes for categorical variables.

### 2.3. Statistical Analysis

Quartile cut-off points of resistin concentrations were derived from controls and applied to the whole study population. We then analyzed the study population characteristics descriptively by case-control status and by quartiles of resistin concentrations. We calculated means (standard deviation (SD)) or medians (and quartiles) depending on the distribution of the variables.

Spearman partial correlation coefficients (and corresponding *p*-values) controlling for age and sex were estimated to assess the correlations between baseline resistin levels and adiposity measurements, other adipokines (adiponectin and leptin), as well as metabolic and inflammatory biomarkers (hsCRP, C-peptide, HDL-C, and ROM) in controls. A correlation coefficient lower than 0.30 was considered as little if any correlation [29].

Conditional logistic regression models were used to estimate RRs and 95% CI of CRC across quartiles of resistin concentrations. Since cases and controls were selected using an incidence density sampling protocol, the odds ratios estimate the incidence rate ratios, which can be interpreted as RRs of the association. Potential confounders were selected as covariates for the models. Higher intakes of fiber and dairy products are established protective factors for CRC, while higher intakes of red meat, processed meat, and alcohol are risk factors [14,15]. Low intakes of vegetables and fish have been associated with a higher risk of CRC [14,15]. Since these variables may also affect resistin concentrations [30], we adjusted for these variables in our regression models. BMI and waist circumference are risk factors for CRC [15], however, the relationship between obesity and resistin is not entirely clear [4,6,31]. Therefore, three models were used to estimate the association. Model 1 was only conditioned on the matching variables. Model 2 was conditioned on matching variables and additionally adjusted for smoking status (never, former, or current smoker), education (none, primary school, technical/professional school, secondary school, or longer education), alcohol abstainers (defined as under 0.3 g/day; yes/no), alcohol consumption (gram/day), physical activity index (inactive, moderately inactive, moderately active, active), energy intake (kcal/day), and dietary intakes of red meat (gram/day), processed meat (grams/day), dietary fiber (grams/day), fruit intake (grams/day), vegetable intake (grams/day), dairy intake (grams/day), and fish and shellfish (grams/day). Model 3 was additionally adjusted for BMI (kg/m^2^) and residuals of BMI-adjusted waist circumference (derived from a regression model with BMI as an independent variable and waist circumference as a dependent variable to avoid multicollinearity). Additional adjustment for height, alcohol intake during lifetime, smoking intensity and duration, or diabetes at baseline (defined as self-reported diabetes diagnosis or HbA1c concentrations ≥ 6.5% at baseline) did not alter the risk estimates appreciably and results are hence not shown. Resistin concentrations were log-transformed to approximate a normal (Gaussian) distribution and included as a continuous variable in conditional logistic regression models [32]. We also analyzed the associations between resistin concentrations and the risk of CRC by modeling restricted cubic polynomial splines with knots at the 5th, 35th, 65th, and 95th percentiles of resistin distribution. We tested for non-linearity using a likelihood ratio test to compare full multivariable-adjusted conditional logistic regression models, including both the linear and cubic spline term, and reduced multivariable-adjusted conditional logistic regression models with only the linear term. The test for non-linearity was not significant in the main analysis (*p* = 0.35) or the subgroups, except for the subgroup of men (*p* = 0.03). Results of non-linearity tests did not change implicitly after excluding participants diagnosed with CRC within 2 years after recruitment. Thus, using both log-transformed and quartile scales was considered sufficient for better capturing the relationship between resistin concentrations and the risk of CRC [32].

The associations between resistin concentrations and the risk of CRC were further assessed according to tumor anatomical subsite (colon, rectum), sex (female, male, as well as with restriction to postmenopausal women [5]), the combination of subsite and sex, by the length of follow-up (≤2 years, 2–5 years, >5 years), BMI (<25, ≥25), hsCRP (<3 mg/L, ≥3 mg/L), baseline diabetes (yes/no), fasting status (≤6 h, >6 h), and C-peptide (<2 ng/mL, ≥2 ng/mL). Sensitivity analyses were carried out by repeating all of the analyses with the exclusion of participants with less than 2 years of follow-up, and participants with extreme resistin levels (defined as concentrations of 1.5 times the interquartile range below the first and above the third quartile [33]). Analyses were also restricted to participants with no missing covariates’ data (complete case analyses).

We finally pooled the relative risk estimate from our main analysis with the relative risk from the published result from the WHI [5] using a random-effects meta-analysis with an inverse variance method. Heterogeneity was assessed using Cochran’s Q-statistic test and inconsistency index (I2).

Minimal detectable RRs for binary exposure were estimated for the second, third, or fourth quartiles as compared to the first quartile in matched case-control using the software “Power” version 2.10 (Channing laboratory, Boston, MA, USA) [34]. With 1293 study participants for each RR estimate, 80% power, and alpha = 0.05, the minimum detectable RR is 1.37 under the assumption of no correlation of exposure in matched pairs, and 1.42 with a recommended correlation of exposure between cases and matched controls of 0.2.

All statistical tests were two-sided, and *p*-values less than 0.05 were considered statistically significant. All analyses were performed using SAS^®^ Enterprise Guide^®^ 8.3 (SAS Institute Inc., Cary, NC, USA) and R version 4.0.5 (R Foundation for Statistical Computing, Vienna, Austria).

## 3. Results

The mean follow-up time between recruitment and CRC diagnosis in cases for the current study was 4.8 ± 2.7 years. Cases and controls had similar distributions of the matching factors (Appendix A). On average, cases at baseline were less physically active, had a higher BMI, a higher waist circumference, and a slightly lower dietary fiber intake compared to controls (Appendix A). Cases also had higher circulating levels of HMW adiponectin, ROM, hsCRP, and HbA1c than controls. Resistin concentrations at baseline were similar in incident CRC cases (4.7 ± 2.0 ng/mL) and controls (4.7 ± 2.2 ng/mL).

The main characteristics of controls in the population according to the quartiles of resistin are presented in Table 1. Participants with higher resistin concentrations were more likely to be women, have a higher age, a lower proportion of university-level education, lower alcohol intake, higher hsCRP, higher C-peptide, and lower HDL-C, while less likely to be in fasting status at blood collection. Interestingly, we observed higher levels of resistin in women compared to men (4.82 ± 2.48 vs. 4.53 ± 1.84 ng/mL, *p* = 0.01), and in participants with less than or equal to 6 h of fasting before blood collection compared to those with more than 6 h of fasting (4.79 ± 2.39 vs. 4.38 ± 1.56 ng/mL, *p* ≤ 0.001).

Among controls, after adjustment for age and sex, resistin concentrations were weakly inversely correlated with adiponectin and HDL-C, and weakly positively correlated with C-peptide and hsCRP (correlation coefficient (r) < 0.15). Resistin was not statistically significantly correlated with BMI, waist circumference, leptin, soluble leptin receptors, reactive oxygen metabolites, or HbA1c concentrations (Table 2).

Resistin concentrations were not statistically significantly associated with the risk of CRC. Thus, compared to quartile one, the RRs in quartiles were, quartile two, 1.11; 95% CI: 0.88–1.39, quartile three, 1.21; 95% CI: 0.97–1.53, and quartile four, 1.15; 95% CI: 0.91–1.46, *p*-trend = 0.41. In the continuous scale, RR for doubling resistin concentrations was 1.11; 95% CI: 0.94–1.30; *p* = 0.22 (Table 3, model 3). No statistically significant relationship between resistin and CRC was observed when the associations were further explored by tumor site (colon or rectum) (Table 3) and right-sided and left-sided colon cancer (Appendix A).

Sex-stratified analyses showed no significant association between resistin and CRC (Table 4). The test for interaction by sex was not statistically significant in all study participants (*p* = 0.86), colon cancer patients and their pairs (*p* = 0.72), and rectal cancer patients and their pairs (*p* = 0.10). However, among men, doubling resistin concentrations were related to 1.53-fold the risk of rectal cancer (95% CI: 1.01–2.33).

In stratified analyses, there was no association between resistin with CRC among persons with BMI < 25 kg/m^2^ or BMI ≥ 25 kg/m^2^ (Appendix A), with or without baseline diabetes, C-peptide ≥ 2 ng/mL or <2 ng/mL (data not shown). Among persons with hsCRP ≥ 3 mg/L, persons with higher compared to those with lower resistin concentrations had higher relative risks of CRC, but these associations were not statistically significant (RR per doubling resistin in persons with hsCRP ≥ 3 mg/L, 1.31; 95% CI: 0.94–1.81). No association was observed for doubling resistin concentrations among individuals with hsCRP < 3 mg/L (Appendix A). We observed an increased risk of CRC associated with resistin among those with more than 6 h of fasting as compared to those with less than 6 h of fasting (RR per doubling, 1.45; 95% CI: 1.03–2.03; *p* = 0.03).

When we restricted the main analysis to cases and matched controls who were diagnosed within the first 2 years of follow-up, we found that participants in the highest as compared to the lowest quartile of resistin concentrations had a 1.97-fold risk of CRC (95% CI: 1.06–3.64; *p*-trend < 0.001); RR per doubling of resistin, 1.44; 95% CI: 0.97–2.12; Appendix A. In contrast, when excluding cases and matched controls that were diagnosed within the first 2 years of follow-up, the RR in the highest versus lowest quartile was 1.05; 95% CI: 0.81–1.37; *p*-trend = 0.79; RR per doubling, 1.03; 95% CI: 0.86–1.24; Appendix A.

The main results were not substantially different when performing complete case analyses or when excluding participants with extreme resistin levels (Appendix A). All findings from subgroup analyses (including analyses by the combination of sex and tumor subsite, by fasting status) were no longer statistically significant when participants who were diagnosed within the first two years of follow-up were excluded (data not shown). In this analysis, the relative risk of rectal cancer per doubling resistin concentrations among men was 1.13; 95% CI: 0.70–1.82.

When we combined our results with those published from the WHI, the pooled RRs of the highest versus the lowest quartile of resistin concentrations were 1.10; 95% CI: 0.93–1.29 for all study participants combined and 1.09; 95% CI: 0.86–1.39 for postmenopausal women. No significant heterogeneity was found (I2 = 0.0%, *p* = 0.77, identical in both meta-analyses).

## 4. Discussion

In this nested case-control study, we found no statistically significant associations between pre-diagnostic circulating resistin concentrations and the risk of CRC. Furthermore, resistin concentrations were not correlated with measures of adiposity and leptin, weakly inversely correlated with adiponectin and HDL-C, and weakly positively correlated with C-peptide and hsCRP.

To our knowledge, the association between circulating resistin and CRC risk has to date been investigated only once previously in a prospective study, within the WHI, which reported a relative risk of 1.04 (95% CI: 0.72–1.50) among postmenopausal women when comparing the highest versus lowest quartile of resistin concentrations [5]. This is consistent with our findings, which extend the evidence to a general European population of men and women. Indeed, we found no heterogeneity when combining our results with those published by the WHI. These data overall suggest that higher pre-diagnostic resistin concentrations are not associated with a higher risk of CRC. In subgroup analyses, we found that higher resistin concentrations were associated with a higher risk of rectal cancer in men and in individuals with more than 6 h of fasting before the blood collection in the current study. However, the results of this subgroup analysis should be interpreted cautiously in light of the multiple tests and small sample sizes for each subgroup. In addition, the association was no longer significant when we excluded participants with less than two years of follow-up. Of note, we observed higher levels of resistin in women compared to men, and in non-fasting compared to fasting participants only in controls, but not in cases. In line with our observations, higher resistin levels in women compared to men have been observed previously [35]; however, except for the non-significant associations in postmenopausal women [5], we are not aware of any comparable studies investigating a relationship between higher resistin levels and the risk of CRC stratifying by sex, tumor site, or the combination of sex and tumor site. Furthermore, in contrast to our findings, several previous studies suggested that circulating resistin levels are not influenced by fasting status [35,36]. Taken together, we cannot rule out the possibility of weak associations among these groups.

Previous case-control studies have reported higher resistin levels in CRC patients (at or after diagnosis) compared to healthy individuals [17,37,38]. Of note, only six (out of thirteen) observational studies reported results from multivariable-adjusted regression models rather than unadjusted case-control comparisons (Appendix A) [5,37,38,39,40,41]. All these studies showed that participants with CRC had higher resistin concentrations than those without CRC. The results of these retrospective studies warrant prudent interpretation because resistin levels were measured in post-diagnostic blood samples; therefore, elevated resistin levels are likely a result of existing tumors [40,42]. Interestingly, we found a significant association between higher resistin concentrations and a higher risk of CRC in participants diagnosed with CRC within two years after enrollment and their matched controls, whereas there was no such association among persons with more than two years of follow-up. Although individuals who reported prevalent cancers at baseline were excluded from our analysis, we speculate whether some individuals may have had undiagnosed cancer at baseline, which may have led to higher resistin concentrations in these individuals. In contrast, the primary results from our study show that pre-diagnostic resistin concentrations are not related to CRC risk.

Resistin was originally described as an adipokine that purportedly links obesity and cancer via inflammation in humans, and insulin resistance in animals [8]. However, in our study, we found no correlation between resistin and BMI as well as waist circumference. Consistent with our findings, previous analyses using different samples from the EPIC-Potsdam study, which were collected at baseline (1994–1998) [43] or between 2010 and 2013 [6], also suggested no such association. Furthermore, we previously performed an analysis on healthy individuals in whom body fat was assessed using magnetic resonance imaging (MRI) scans [6]. In that study, subcutaneous and visceral adipose tissue accounted for only 1% of the explained variance in plasma resistin concentrations [6]. Several previous studies, including population-based studies, reported comparably weak (r < 0.2) but statistically significant correlations between resistin concentrations and BMI [6,30,44], or waist circumference [5,6,44]. Taken together, the current evidence suggests that there is no substantial correlation between resistin and BMI or waist circumference in humans. Our current study confirmed findings from other studies that showed that resistin was also not substantially correlated with other adipokines secreted by adipose tissues, such as adiponectin or leptin (r < 0.15) [5]. These findings strengthen the point that the resistin–obesity relationship is different in humans and animals, and do not support the role of circulating resistin as a biomarker of adipose tissue mass in humans.

We found null or weak correlations between resistin and other metabolic and inflammatory biomarkers among controls, which are consistent with most results published in population-based studies. The findings are consistent with previous findings from cross-sectional studies regarding HbA1c [30,45], HDL-C [44], C-peptide [30,45], and hsCRP [30,45]. In addition, previous studies found that resistin was weakly correlated with other inflammatory molecules such as TNF-α, IL-6, insulin [5,30], and insulin resistance measured by homeostasis model assessment-insulin resistance (HOMA-IR) [46]. However, in a small retrospective case-control study (40 CRC cases, 40 controls), significant correlations between resistin and inflammatory and metabolic biomarkers such as hsCRP and HDL-C were found in CRC patients, but not in the control group [40]. Furthermore, we observed a null association between resistin concentrations and CRC risk in subgroup analyses by BMI, hsCRP, C-peptide, and baseline diabetes. Taken together, based on these lines of evidence we speculate that high levels of resistin in CRC patients are likely a result of existing tumors being accommodated by inflammation.

The current study has several strengths, including its prospective design, long follow-up time before CRC diagnosis, and a large number of cases. The current study included participants from several European countries, which improves the generalizability of the results. The current study had some limitations. First, resistin was measured in blood samples that had been stored over a longer period of time. The stability of resistin measured in frozen blood samples, however, was confirmed by a high overall intraclass correlation (0.95) over samples stored for 4 years, 2 years, and 1 year at −70 °C, suggesting that resistin levels may not be affected by long-term frozen storage [47]. The measured values of resistin in our data are consistent with those of other studies using the same assay (ELISA) and vendor (BioVendor) in similar study participants [6] or controls [41]. Second, a single measurement of biomarker levels at baseline may not reflect the long-term exposure to biomarkers. However, a previous study showed that there was no significant difference but high reliability (intraclass correlation coefficient 0.70) between resistin concentrations in blood samples collected at baseline and one year after [36], which suggested that a single measurement of circulating resistin is reasonable to represent long-term exposure. Third, although we included only CRC incident cases in our study, we cannot exclude the possibility that some participants had existing but as yet undiagnosed cancer at the time of recruitment. To overcome this issue, we performed analyses excluding participants with a follow-up time of less than two years and the results did not change appreciably. Fourth, as a collaborative endeavor from 10 European countries, some assessment and follow-up methods as well as blood sample storage methods differed across centers in the EPIC study. A calibration study as well as a comprehensive re-coding scheme were applied to standardize the data [20]. Furthermore, it has been shown that CRC case ascertainment was not different between passive and active follow-up methods [48]. We found no significant inter-assay variation of resistin concentrations between plates. Thus, these subtle differences are unlikely to have impacted our analysis.

## 5. Conclusions

In conclusion, higher pre-diagnostic resistin concentrations were not associated with a higher risk of CRC in men and women. Furthermore, our study suggests that resistin concentrations are null or weakly correlated with general or abdominal obesity measures, or metabolic and inflammatory biomarkers. Our findings suggest that pre-diagnostic circulating resistin concentrations are not associated with CRC risk, however, the presence of weak associations cannot be ruled out.

## Figures and Tables

**Table 1 cancers-14-05499-t001:** Characteristics of controls at baseline (n = 1293), by quartiles of resistin concentration in the nested case-control study, European Prospective Investigation into Cancer and Nutrition, 1992–2005.

	Quartiles of Resistin Concentration			
	Q1 (N = 324)	Q2 (N = 325)	Q3 (N = 321)	Q4 (N = 323)
Resistin quartile ranges (ng/mL)	≤3.47	3.47< to ≤4.28	4.28< to ≤5.42	5.42< to ≤34.41
Age at blood collection, years, mean (SD) ^a^	57.6 (6.9)	57.5 (7.2)	58.1 (6.7)	59 (7.1)
Women, n (%) ^a^	154 (47.5)	183 (56.3)	157 (48.9)	187 (57.9)
Postmenopausal women, baseline, n (%) ^a^	123 (38.0)	135 (41.5)	116 (36.1)	141 (43.7)
Fasting (>6 h), n (%) ^a^	98 (30.2)	98 (30.2)	74 (23.1)	80 (24.8)
BMI, kg/m^2^, mean (SD)	26.2 (3.5)	26.6 (4.1)	26.3 (3.8)	26.2 (3.7)
Waist circumference, cm, mean (SD) ^b^	89.2 (12.5)	88.4 (12.2)	88.8 (11.8)	87.7 (12.6)
Diagnosed diabetes at baseline (self-reported or HbA1C ≥ 6.5%), n (%)	19 (5.9)	18 (5.5)	13 (4.0)	14 (4.3)
Current smoker, n (%)	84 (25.9)	86 (26.5)	68 (21.2)	85 (26.3)
University degree or higher education level, n (%)	68 (21)	52 (16)	51 (15.9)	58 (18)
Physically moderately active, or active, sex-specific, n (%)	178 (54.9)	176 (54.2)	168 (52.3)	195 (60.4)
Alcohol abstainers (<0.3 g/day), n (%)	38 (11.7)	45 (13.8)	43 (13.4)	51 (15.8)
Alcohol consumption, g/day, median (Q1–Q3) ^b^	10.6 (3.2–28.1)	8.5 (1.6–23.5)	7.7 (1.7–19.9)	6.0 (1.0–16.8)
Energy intake, Kcal/day, median (Q1–Q3) ^b^	2084 (1608–2478)	2010 (1638–2404)	2052 (1719–2511)	1997 (1590–2464)
Red meat, g/day, median (Q1–Q3) ^b^	43.4 (24.4–75.3)	43.0 (25.3–69.9)	49.6 (27.4–76.5)	47.7 (24.2–74.9)
Processed meat, g/day, median (Q1–Q3) ^b^	24.6 (13.0–42.6)	23.3 (12.5–45.9)	28.4 (16.8–44.3)	23.2 (12.9–42.6)
Dietary fiber, g/day, median (Q1–Q3) ^b^	23.6 (18.0–28.6)	22.8 (18.0–27.5)	23.2 (18.0–27.8)	22.1 (17.9–26.7)
Fruit intake, g/day, median (Q1–Q3) ^b^	191.5 (106.4–314.3)	195.2 (108.2–330.2)	176.9 (98.2–279.8)	191.5 (109.3–318.4)
Vegetable intake, g/day, median (Q1–Q3) ^b^	160.9 (104.3–230.8)	166.8 (103.8–248.1)	148.1 (96.0–221.6)	157.9 (97.4–241.5)
Dairy intake, g/day, median (Q1–Q3) ^b^	278.9 (155.2–435.3)	310.6 (175.0–451.0)	318.2 (171.2–495.9)	305.9 (161.5–484.5)
Fish and shellfish, g/day, median (Q1–Q3) ^b^	29.0 (13.4–52.3)	29.7 (16.4–51.8)	29.4 (14.2–49.3)	30.0 (13.6–52.4)
Adiponectin, µg/mL, median (Q1–Q3) ^b^	7.3 (5.2–10.1)	7.3 (5.5–9.7)	6.9 (5.0–9.2)	7.2 (5.3–9.6)
HMW adiponectin, µg/mL (Q1–Q3) ^b^	3.7 (2.4–5.7)	3.8 (2.4–5.4)	3.6 (2.2–5.1)	3.7 (2.4–5.2)
Leptin, ng/mL (Q1–Q3) ^b^	6.3 (3.4–14.1)	9.4 (5.1–19.2)	7.7 (3.5–16.6)	6.9 (3.6–17.0)
Soluble leptin receptor, ng/mL, mean (SD) ^b^	23 (8.9)	21.9 (7.9)	22.7 (8.6)	23.6 (15.8)
ROM, Carratelli units, mean (SD) ^b^	381 (71.8)	382.4 (69.4)	379.1 (64.2)	397.4 (71.7)
hsCRP, mg/L, median (Q1–Q3) ^b^	1.9 (0.8–3.7)	2.2 (0.9–4.0)	2.1 (0.9–4.2)	2.9 (1.3–5.8)
C-peptide, ng/mL, median (Q1–Q3) ^b^	3.6 (2.5–5.9)	3.7 (2.6–5.6)	4.0 (2.9–6.3)	4.3 (2.9–5.9)
HDL-C, mmol/L, median (Q1–Q3) ^b^	1.5 (1.2–1.8)	1.5 (1.2–1.7)	1.4 (1.2–1.7)	1.4 (1.1–1.7)
HbA1c, %, mean (SD) ^b^	5.8 (0.6)	5.8 (0.8)	5.7 (0.6)	5.7 (0.5)

^a^ Matching variable. ^b^ Among users only. Abbreviations: Q1–Q3, the first quartile (the 25th percentile) to the third quartile (the 75th percentile); SD, standard deviation; BMI: body mass index; HMW, high-molecular weight; ROM, reactive oxygen metabolites; HDL-C, high-density lipoprotein cholesterol (HDL-C); hsCRP, high-sensitivity C-reactive protein; HbA1c, glycated hemoglobin A1c.

**Table 2 cancers-14-05499-t002:** Age- and sex-adjusted Spearman partial rank correlations between resistin and adiposity measurements, other adipokines, as well as metabolic and inflammatory biomarkers in controls (n = 1293), the European Prospective Investigation into Cancer and Nutrition Cohort (1992–2005).

	Number of Participants	Correlation Coefficient (r)	*p*-Value
BMI, kg/m^2^	1293	−0.02	0.52
Waist circumference, cm	1216	−0.03	0.28
Adiponectin, µg/mL	651	−0.08	0.04
HMW adiponectin, µg/mL	650	−0.07	0.08
Leptin, ng/mL	651	−0.002	0.96
Soluble leptin receptor, ng/mL	651	−0.02	0.56
ROM, Carratelli units	723	0.07	0.05
CRP-hs, mg/L	727	0.12	<0.01
C-peptide, ng/mL	622	0.09	0.02
HDL-C, mmol/L	726	−0.12	<0.01
HbA1c, %	606	−0.05	0.24

r: correlation coefficient; BMI, body mass index; HMW, high-molecular weight; ROM, reactive oxygen metabolites; HDL-C, high-density lipoprotein cholesterol (HDL-C); hsCRP, high-sensitivity C-reactive protein; HbA1c, glycated hemoglobin A1c.

**Table 3 cancers-14-05499-t003:** Relative risk (RR) and 95% confidence interval (95% CI) estimated for the association between resistin concentrations and the risk of colorectal cancer in the EPIC study data (1992–2005) in conditional logistic regression models.

	Quartile Form					Continuous Form	
	Q1	Q2	Q3	Q4	*p*-Trend ^a^	Doubling Resistin Concentrations ^b^	*p*-Value
**Resistin quartile ranges (ng/mL)**	≤3.47	3.47< to ≤4.28	4.28< to ≤5.42	5.42< to ≤34.41			
Colorectal Cancer							
No. cases/controls	297/324	317/325	348/321	331/323		1293/1293	
Model 1	ref	1.06 (0.85–1.32)	1.19 (0.95–1.49)	1.13 (0.90–1.41)	0.46	1.11 (0.95–1.30)	0.19
Model 2	ref	1.10 (0.88–1.37)	1.22 (0.97–1.53)	1.15 (0.91–1.46)	0.38	1.12 (0.95–1.31)	0.18
Model 3	ref	1.11 (0.88–1.39)	1.21 (0.97–1.53)	1.15 (0.91–1.46)	0.41	1.11 (0.94–1.30)	0.22
Colon Cancer							
No. cases/controls	165/174	178/193	203/188	211/202		757/757	
Model 1	ref	0.97 (0.73–1.30)	1.16 (0.86–1.56)	1.11 (0.83–1.50)	0.62	1.14 (0.94–1.40)	0.19
Model 2	ref	1.01 (0.75–1.37)	1.17 (0.86–1.59)	1.15 (0.84–1.56)	0.67	1.15 (0.93–1.42)	0.19
Model 3	ref	1.04 (0.76–1.41)	1.20 (0.88–1.65)	1.20 (0.88–1.65)	0.53	1.18 (0.95–1.47)	0.13
Rectal Cancer							
No. cases/controls	120/134	119/115	127/118	109/108		475/475	
Model 1	ref	1.15 (0.81–1.64)	1.20 (0.85–1.71)	1.13 (0.78–1.64)	0.76	1.07 (0.82–1.39)	0.62
Model 2	ref	1.17 (0.81–1.68)	1.26 (0.88–1.82)	1.20 (0.81–1.76)	0.64	1.09 (0.83–1.43)	0.55
Model 3	ref	1.16 (0.80–1.68)	1.25 (0.86–1.80)	1.17 (0.79–1.73)	0.69	1.06 (0.80–1.41)	0.66

Model 1: Conditioned on matching factors only: age, sex, study center, time of the day at blood collection, and fasting status. Women were further matched by menopausal status, phase of the menstrual cycle, and use of oral contraceptives at blood collection, and postmenopausal women were matched by hormone replacement therapy use. Model 2: Model 1 + smoking status, education, alcohol consumption, alcohol abstainers, physical activity index, energy intake, red meat, processed meat, dietary fiber, fruit intake, vegetable intake, dairy intake, fish, and shellfish intake. Model 3: Model 2 + body mass index (BMI), and residuals of BMI-adjusted waist circumference. ^a^
*p*-values for trend derived from models with the median resistin concentration within quartiles as a continuous variable. ^b^ Models with continuous log-transformed resistin concentrations by log 2.

**Table 4 cancers-14-05499-t004:** Relative risk (RR) and 95% confidence interval (95% CI) estimated for the association between resistin concentrations and the risk of colorectal cancer stratified by sex and subsite in the EPIC study data (1992–2005) in conditional logistic regression models.

	Quartile Form					Continuous Form	
	Q1	Q2	Q3	Q4	*p*-Trend ^a^	Doubling Resistin Concentrations ^b^	*p*-Value
Resistin quartile ranges (ng/mL)	≤3.47	3.47< to ≤4.28	4.28< to ≤5.42	5.42< to ≤34.41			
Sex							
Women							
No. cases/controls	134/154	165/183	190/157	192/187		681/681	
RR (95% CI)	ref	1.02 (0.75–1.40)	1.39 (1.00–1.94)	1.21 (0.87–1.67)	0.17	1.06 (0.85–1.33)	0.59
Postmenopausal women							
No. cases/controls	97/120	123/129	136/110	135/132		491/491	
RR (95% CI)	ref	1.14 (0.79–1.63)	1.48 (1.01–2.18)	1.25 (0.85–1.84)	0.25	1.12 (0.86–1.47)	0.38
Men							
No. cases/controls	163/170	152/142	158/164	139/136		612/612	
RR (95% CI)	ref	1.20 (0.85–1.69)	1.05 (0.76–1.47)	1.08 (0.76–1.54)	0.77	1.14 (0.89–1.46)	0.29
Sex and Tumor subsite							
Colon cancer women							
No. cases/controls	77/93	99/108	120/103	133/125		429/429	
RR (95% CI)	ref	1.11 (0.73–1.68)	1.40 (0.91–2.16)	1.37 (0.89–2.11)	0.37	1.17 (0.88–1.57)	0.28
Colon cancer men							
No. cases/controls	88/81	79/85	83/85	78/77		328/328	
RR (95% CI)	ref	0.96 (0.59–1.55)	0.97 (0.60–1.59)	0.95 (0.58–1.57)	1.00	1.13 (0.80–1.58)	0.50
Rectal cancer women							
No. cases/controls	54/53	55/65	62/50	53/56		224/224	
RR (95% CI)	ref	0.70 (0.41–1.19)	1.11 (0.62–2.00)	0.72 (0.40–1.31)	0.29	0.71 (0.46–1.08)	0.11
Rectal cancer men							
No. cases/controls	66/81	64/50	65/68	56/52		251/251	
RR (95% CI)	ref	1.89 (1.06–3.36)	1.43 (0.85–2.40)	1.8 (0.99–3.24)	0.12	1.53 (1.01–2.33)	0.05

Results were based on conditional logistic regression models conditioned on matching factors (age, sex, study center, time of the day at blood collection, and fasting status, women were further matched by menopausal status, phase of the menstrual cycle, and use of oral contraceptives at blood collection, and postmenopausal women were matched by hormone replacement therapy use) and adjusted for smoking status, education, alcohol consumption, alcohol abstainers, physical activity index, energy intake, red meat, processed meat, dietary fiber, fruit intake, vegetable intake, dairy intake, fish and shellfish intake, body mass index (BMI), and residuals of BMI-adjusted waist circumference. ^a^
*p*-values for trend derived from models with the median resistin concentration within quartiles as continuous variables. ^b^ Models with continuous log-transformed resistin concentrations by log 2.

## Data Availability

Raw data cannot be made freely available because of restrictions imposed by the Ethical Committee that do not allow open/public sharing of data of individuals. However, aggregated data are available for other researchers upon request. Requests should be sent to Tobias Pischon (tobias.pischon@mdc-berlin.de).

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
