# Peer review of "Pre-Diagnostic Circulating Resistin Concentrations Are Not Associated with Colorectal Cancer Risk in the European Prospective Investigation into Cancer and Nutrition Study"

_cancers, 2022, doi:10.3390/cancers14225499_

Round 1
Reviewer 1 Report
The rationale for looking into resistin is very questionable. Authors have provided information that resistin is produced by macrophages and monocytes. Please specify the particular polarisation in macrophages, time of the differentiation or whether it is a general marker of those cells. In addition, authors are looking into correlation between resisting and gender, subsite. It is a very broad and perhaps even biased goal of research. Since samples and other clinical data has been collected please choose an endpoint which is more likely to be associated with resistin. Also, most CRC patients are prone to cachexia in advanced stages, thus not sure why the obesity was the point of discssion in this paper. In Johns Hopkins we see that 80-82% of patients are with BMI <20 at primary admission. Obviously, depends on community screening but still it is an issue rather than obesity.
In introduction authors mention that resistin induces expression of inflammatory cytokines. What cytokines, what pathways do they regulate/activate. Please specify.
The epidemiology regarding CRC is very broad. Not in all US states, as example, it is not the second most fatal cancer
CRC cancer, define the MSI status and define whether it is left or right sided cancer.
Do not use abbreviations in subtitles.
Samples collection described in the methods sounds very incoherent. Define reasons why for some countries/centers you do or not collect samples/data etc. What factors defined such discrepancy in methodology. For example, US FDA recommends that trials must follow one scheme for trial conduction etc. Please find info on the FDA website.
Statistical analysis should b reviewed by professional statistician, however in my opinion it should be reduced in the main text and be written succinctly. The detailed data should be presented in supplementary.
Table 1. Please wrap text in the column characteristic. Please amend.
Data should be presented in median + 95% or 99% CI rather than mean numbers. Means are less reliable in clinical trials.
Author Response
We sincerely appreciate your comment. Please see the attachment.

Reviewer 2 Report
The present manuscript is an interesting multicenter prospective study with a long follow-up time and a large number of cases. The topic is interesting, the article is well written.
Author Response
We sincerely appreciate your comments.
Reviewer 3 Report
Resistin is a pro-inflammatory cytokine and is associated with various cellular and metabolic functions. Resistin belongs to adipogenic factors such as PPARs, SREBP-1, SCD, etc. In addition, numerous studies suggest that resistin plays a key role in proliferation, metastasis, angiogenesis and inflammation, as well as in the regulation of tumor cell metabolism. In humans, resistin is mainly secreted from macrophages, whereas in rodents its main source is adipocytes. The normal physiological range of resistin in human serum is 6-25 ng/ml. A variety of studies have shown high serum resistin levels in patients with breast cancer, lymphoma, esophageal squamous cell carcinoma, endometrial adenocarcinoma, gastric cancer and colorectal cancer.
Phan et al. have been focused on concentration of resistin in plasma CRC patients (N=1640). Authors correlated it with several patient characteristics summarized at Table 1. Please, the left part of the table is missing. The level of resistin had been detected in patient serum samples by ELISA and were compared with control samples. I appreciate the distribution of patient sample collection into 4 groups (Q1-4). The authors have found that higher resistin concentrations is in correlation with a higher risk of rectal cancer in men in this extend study (Table 4). However, concentration of resistin was not significantly associated with risk of CRC in whole population (Table 3). Despite large number of patient samples the p values were only slightly significant. Clinical studies have shown the presence of high serum resistin levels in patients with various types of tumours. Evidence suggests that resistin could be a potential biomarker in cancer. This hypothesis was not confirmed by the authors. This is an important report for another studies and therefore I support the paper submission.
Author Response
We sincerely appreciate your comments. We modified the margins of table 1.